# Tipping the Balance: Vitamin D Inadequacy in Children Impacts the Major Gut Bacterial Phyla

**DOI:** 10.3390/biomedicines10020278

**Published:** 2022-01-26

**Authors:** Parul Singh, Arun Rawat, Marwa Saadaoui, Duaa Elhag, Sara Tomei, Mohammed Elanbari, Anthony K. Akobeng, Amira Mustafa, Ibtihal Abdelgadir, Sharda Udassi, Mohammed A. Hendaus, Souhaila Al Khodor

**Affiliations:** 1Research Department, Sidra Medicine, Doha P.O. Box 26999, Qatar; psingh@sidra.org (P.S.); arawat@sidra.org (A.R.); msaadaoui@sidra.org (M.S.); delhag@sidra.org (D.E.); stomei@sidra.org (S.T.); melanbari@sidra.org (M.E.); 2College of Health and Life Sciences, Hamad Bin Khalifa University, Doha P.O. Box 5825, Qatar; 3Division of Gastroenterology, Hepatology, and Nutrition, Sidra Medicine, Doha P.O. Box 26999, Qatar; aakobeng@sidra.org; 4Pediatric Department, Sidra Medicine, Doha P.O. Box 26999, Qatar; amustafa@sidra.org (A.M.); sudassi@sidra.org (S.U.); mrahal@sidra.org (M.A.H.); 5Emergency Department, Sidra Medicine, Doha P.O. Box 26999, Qatar; iabdelgadir@sidra.org

**Keywords:** pediatric vitamin D deficiency, host genetics, gut microbiota, Qatar, *Bacteroidetes* to *Firmicutes* ratio

## Abstract

Vitamin D inadequacy appears to be on the rise globally, and it has been linked to an increased risk of osteoporosis, as well as metabolic, cardiovascular, and autoimmune diseases. Vitamin D concentrations are partially determined by genetic factors. Specific single nucleotide polymorphisms (SNPs) in genes involved in vitamin D transport, metabolism, or binding have been found to be associated with its serum concentration, and these SNPs differ among ethnicities. Vitamin D has also been suggested to be a regulator of the gut microbiota and vitamin D deficiency as the possible cause of gut microbial dysbiosis and inflammation. This pilot study aims to fill the gap in our understanding of the prevalence, cause, and implications of vitamin D inadequacy in a pediatric population residing in Qatar. Blood and fecal samples were collected from healthy subjects aged 4–14 years. Blood was used to measure serum metabolite of vitamin D, 25-hydroxycholecalciferol 25(OH)D. To evaluate the composition of the gut microbiota, fecal samples were subjected to 16S rRNA gene sequencing. High levels of vitamin D deficiency/insufficiency were observed in our cohort with 97% of the subjects falling into the inadequate category (with serum 25(OH)D < 75 nmol/L). The CT genotype in rs12512631, an SNP in the *GC* gene, was associated with low serum levels of vitamin D (ANOVA, *p* = 0.0356) and was abundant in deficient compared to non-deficient subjects. Overall gut microbial community structure was significantly different between the deficient (D) and non-deficient (ND) groups (Bray Curtis dissimilarity *p* = 0.049), with deficient subjects also displaying reduced gut microbial diversity. Significant differences were observed among the two major gut phyla, *Firmicutes* (F) and *Bacteroidetes* (B), where deficient subjects displayed a higher B/F ratio (*p* = 0.0097) compared to ND. Vitamin D deficient children also demonstrated gut enterotypes dominated by the genus *Prevotella* as opposed to *Bacteroides*. Our findings suggest that pediatric vitamin D inadequacy significantly impacts the gut microbiota. We also highlight the importance of considering host genetics and baseline gut microbiome composition in interpreting the clinical outcomes related to vitamin D deficiency as well as designing better personalized strategies for therapeutic interventions.

## 1. Introduction

Vitamin D and its metabolites play a crucial role in early life, including bone growth [1] and development of the immune system [2]. Recent epidemiologic reports linking low vitamin D levels in children to diabetes, metabolic syndrome, asthma, dermatitis, and anemia have piqued interest in pediatric vitamin D investigations [3]. Even though the cutoff of vitamin D deficiency and insufficiency varies between institutions, with some recommending levels above 75 nmol/L as sufficient or normal [4], the consensus is that ideal 25(OH)D levels should be greater than 50 nmol/L [5]. This limit was based on data indicating levels less than 50 nmol/L are linked to a variety of disease outcomes [6].

In the Middle East, low levels of serum vitamin D have been seen in people of all ages and genders [7], and vitamin D inadequacy (below 75 nmol/L) has been widely reported in the GCC (Gulf cooperative council) populations, particularly among children [8,9]. In a study of 331 Saudi children aged 6 to 17 years, 71.6% had vitamin D levels below 50 nmol/L [10]. Similarly, in a cohort of 293 adolescent girls (11–18 years) living in the United Arab Emirates, the authors reported vitamin D deficiency (<27.5 nmol/L) in 78.8% of the study subjects [11]. In Qatar, 61% (11–16 years old) adolescents, 29% (5–10 years old) children, 9.5% (below 5 years old) children demonstrated serum 25(OH)D levels <75 nmol/L, along with delayed milestones, gastroenteritis, fractures, and rickets [12].

Vitamin D levels are affected by skin pigmentation, sun protection, latitude, age, and exposure to sunlight [13]. Studies have also shown that genetic factors play a significant role in determining the serum 25(OH)D levels in various populations such as Caucasians, African American, Arabs, and Asian [14,15,16,17]. These studies have shown that single nucleotide polymorphism (SNPs) in several candidate genes involved in vitamin D metabolism are associated with its low levels and that the allele frequencies of some of these SNPs vary between populations [14,18]. Despite the widespread prevalence of vitamin D deficiency in children living in Qatar [12,19], there has yet to be a study examining whether a genetic susceptibility to vitamin D deficiency exists in this multi-ethnic population.

The link between vitamin D and the composition of the gut microbiome is well established [20,21]. More than 2000 species of bacteria comprise the gut microbiota and are distributed throughout the gastrointestinal tract (GIT) [22]. The gut microbiota is responsible for a variety of tasks, including nutrition metabolism, vitamins synthesis, and short chain fatty acid (SCFA) generation, tight junction barrier function and modulation, antimicrobial agent release, and immunological regulation [23,24,25]. Several studies suggest that both vitamin D status and supplementation has an impact on the composition of the gut microbiome [21,26,27,28]. Vitamin D intake was found to be inversely associated with *Prevotella* abundance and positively associated with *Bacteroides* abundance in a cross-sectional study of healthy subjects [29]. Another study conducted with healthy, but overweight or obese individuals found a higher abundance of genus *Coprococcus* and a lower abundance of genus *Ruminococcus* in the study participants with sufficient serum 25(OH)D (>75 nmol/L) as compared to those with lower serum 25(OH)D (<50 nmol/L) [30]. Vitamin D supplementation dramatically boosted the gut microbial diversity and abundance of probiotic taxa like *Akkermansia* and *Bifidobacterium*, according to data from our recent study on a cohort of healthy females [21]. Dynamic shifts of genera *Bacteroides* and *Prevotella* were also noted, indicating a change in intestinal enterotypes following supplementation [21]. The evidence presented above supports the idea that vitamin D metabolism is inextricably linked to human gut microbiota composition. A recent study done in older men also showed that the gut microbiome influences the level of active vitamin D and its metabolism [20], wherein the phylum Firmicutes was positively linked to increased amounts of the active form of vitamin D (1,25(OH)2D) [20]. Such studies, however, are lacking in the pediatric population despite the vital role of vitamin D in early stages of life.

The goal of this study is to gain better understating of the association between vitamin D levels, host genetics, and gut microbiota composition in a pediatric population living in Qatar. Improving our knowledge of this complex interaction is crucial for delineating the determinants of the vitamin D status as well as for stratifying subjects for supplementation and early personalized interventions.

## 2. Methods

### 2.1. Study Participants and Design

The approval for the study was obtained from Sidra Medicine IRB (1708012909). The study followed the latest iteration of the Declaration of Helsinki as well as the ICH Guidelines for Good Clinical Practice (CPMP/ICH/135/95), published in July 1996. Children who visited Sidra (Pediatric clinics and the Emergency department) were evaluated for study eligibility and enrolled. Before being included in the study, all participants underwent a physical examination and gave their informed consent. All the participants were in good health and had no underlying disorders or chronic conditions. They had not taken vitamin D in the previous 6 months and had not been exposed to antibiotics in the previous month. All participants were requested to fill out a questionnaire about their current and previous medical histories, supplementation information, dietary habits, sun exposure, and other pertinent information.

### 2.2. Vitamin D Serum Measurement

Around 3.5 mL of blood was collected in a serum separator tube (SST). Vitamin D serum levels were tested at the Sidra Medicine Pathology laboratory using in a two-step competitive binding immunoenzymatic assay. In the initial incubation, samples were added to a reaction vessel with a vitamin D-binding protein (VDBP) releasing agent and paramagnetic particles coated with a sheep monoclonal anti-25(OH) vitamin D antibody; 25(OH) vitamin D is released from VDBP and binds to the immobilized monoclonal anti-25(OH) vitamin D in the solid phase. Subsequently, a 25(OH) vitamin D analogue, alkaline phosphatase conjugate, was added, which competes for binding to the immobilized monoclonal anti-25(OH) vitamin D. After a second incubation, materials bound to the solid phase were held in a magnetic field while unbound materials were washed away. Then, the chemiluminescent substrate Lumi-Phos* 530 (Lumigen, Southfield, MI, USA) was added to the vessel, and light generated by the reaction was measured with a luminometer. The light production was inversely proportional to the concentration of 25(OH) vitamin D in the sample. The amount of analytes in the sample was determined from a stored, multi-point calibration curve. Although there is no unanimity on the required serum levels of 25(OH)D, the Endocrine Society has defined the levels below a threshold of 50 nmol/L (or 20 ng/mL) as vitamin D deficient [31]. Furthermore, various expert bodies and societies have established 50 nmol/L as the “vitamin D requirement of almost all normal healthy adults,” using bone health as the primary criterion. In their “Dietary Reference Intakes” the Institute of Medicine (IOM, Washington, DC, USA) proposes a threshold level of 50 nmol/L [32]. According to some evidence, a 25(OH)D level greater than 50 nmol/L may be necessary for effective risk reduction for a variety of outcomes [33], which is also supported by the main pediatric societies such as American Academy of Pediatrics (AAP), European Society for Pediatric Gastroenterology Hepatology and Nutrition (ESPGHAN) and the Committee on Nutrition of the Spanish Association of Pediatrics (AEP) [34]. Based on the above criteria, the participants were classified as either deficient (those with serum levels of 25(OH)D below <50 nmol/L) or non-deficient (those with serum levels of 25(OH)D above >50 nmol/L) [35,36].

### 2.3. Genotyping

Based on an exhaustive literature review, we chose 37 SNPs from 6 candidate genes that met the following criteria: (1) biological significance in vitamin D metabolism, transport, or degradation; and (2) previous GWAS evidence of a substantial association. [37]. The selected genes were group-specific component (*GC*, coding for vitamin D binding protein VDBP), vitamin D receptor (*VDR*), vitamin D 25-hydroxylase (*CYP2R1*), 1-alpha-hydroxylase (*CYP27B1*), vitamin D 24-hydroxylase (*CYP24A1*), and 7-dehydrocholesterol reductase (*DHCR7*/*NADSYN1*). The roles of these selected genes in the vitamin D metabolic cascade are shown in Figure 1. Genotyping was performed by high-throughput quantitative PCR (qPCR), using the Fluidigm Biomark HD platform (Fluidigm Corporation, South San Francisco, CA, USA), as previously described [38]. Samples were run in duplicate. Genotyping calls were assessed based on the allele discrimination plots and single amplification plots. The genotyping calls were saved as .csv files and processed for further investigation. The replicates of SNP rs757343 produced discordant calls and were thus removed from the analysis. Hardy–Weinberg equilibrium (HWE) for each SNP was tested using the chi-square test. SNPs rs10877012 and rs7041 did not meet HWE, and SNP rs2882679 gave one genotype only; thus, they were also discarded from further analysis. The chi-square test was used to test the association with D/ND classification. An ANOVA test was performed to test the association between genotypes and vitamin D levels. *p*-value < 0.05 was considered significant for all statistical assessments.

### 2.4. Microbial DNA Extraction from Stool Samples

A portion (400 mg) of the stool sample was transferred to the OMNIgene GUT kit (DNA Genotek Inc, Ottawa, Canada). Microbial DNA was extracted using the QIAamp Fast DNA Stool Mini Kit. In a 2 mL tube, 200 mg of fecal sample was combined with 0.5 mL of InhibitEX buffer and vortexed until well homogenized. Thereafter, the samples were combined with 0.2 g of sterile zirconia/silica beads (diameter, 0.1 mm; Biospec Product, ROTH, Karlsruhe, Germany), vortexed, and incubated at 70 °C for 10 min to finish the lysis. The supernatant (600 mL) was transferred into a 2.0 mL microcentrifuge tube containing 25 mL proteinase K. The subsequent steps were carried out as per the instruction of the QIAamp DNA stool MiniKit. The eluted DNA samples (50 μL) were stored at −20 °C until library preparation.

### 2.5. Bacterial 16S rRNA PCR Amplification and High Throughput Sequencing

Polymerase chain reaction (PCR) was used to amplify the 16S rRNA variable regions V3 and V4 using the Illumina suggested amplicon primers with adapters.

Forward:

5′ TCGTCGGCAGCGTCAGATGTGTATAAGAGACAGCCTACGGGNGGCWGCAG ’3.

Reverse: 5′GTCTCGTGGGCTCGGAGATGTGTATAAGAGACAGGACTACHVGGGTATCTAATCC ’3.

In a 25 μL reaction mixture, 5 μL of each forward and reverse primer (1 μM), 2.5 μL of template DNA, and 12.5 μL of 1× Phusion Hot start Master Mix (Thermo scientific, Waltham, MA, USA) were combined. The amplifications were carried out using a Thermo Scientific Veriti 96-well thermal cycler using the following program: initial denaturation at 95 °C for 2 min, followed by 30 cycles of denaturation at 95 °C for 30 s, primer annealing at 60 °C for 30 s, and extension at 72 °C for 30 s, with a final elongation at 72 °C for 5 min used for the amplifications. Amplicons were then purified according to the Illumina MiSeq 16S Metagenomic Sequencing Library Preparation protocol on 27 January 2020 (http://support.illumina.com/downloads/16s_metagenomic_sequencing_library_preparation.html). Samples were multiplexed using the Nextera XT Index kit (Illumina, San Diego, CA, USA) according to the manufacturer’s instructions. The amplicons were then pooled to achieve an equimolar library concentration. Illumina MiSeq platform (Illumina, San Diego, CA, USA), at the Sidra research facility, was used for sequencing of the final pooled product using a MiSeq Reagent Kit v3 (paired-end 2 × 300 bp).

### 2.6. 16S Sequence Data Processing and Analysis

Fast QC (http://www.bioinformatics.babraham.ac.uk/projects/fastqc) was used to assess the sequencing quality on 2 October 2020. The Quantitative Insights into Microbial Ecology (QIIME2; version 2019.4.0) software package [39,40] was used to input the demultiplexed sequencing data. Even though the overall distribution was consistent, samples 24, 26, 38, and 48 had poor sampling depth and were excluded from the final analysis. The rarefaction curves tapered phylogenetically, showing that the complete microbial population was sufficiently represented, and the samples were rarefied at a depth of ≥3531. The data were denoised with DADA2 [41]. Taxonomic classification was performed utilizing Silva [42] for QIIME2 classifier (version silva-132-99-515-806). The data were then imported into R (RStudio v 1.4 with R v 4.1) [43] in a Biological Observation Matrix (biom.) format, before further evaluation with the Phyloseq package [44].

Observed species, Chao1 [45] Shannon [46], and inverse Simpson (InvSimpson) [47] indices were used to quantify alpha diversity in RStudio using the R package “vegan” (v2.5–6) [48]. Weighted Unifrac, Unweighted Unifrac, Bray–Curtis, and Jaccard distance metrics were used to measure beta diversity. Bray–Curtis dissimilarity was employed to establish significance, and CCA was utilized as an ordination approach. We also used Wilcoxon or Kruskal–Wallis nonparametric statistical tests, followed by Dunn post hoc analysis when needed. Bonferroni correction was used to compute the false discovery rate (FDR), with a *p*-value of 0.05 considered significant for all tests.

Metagenome functional contents were analyzed using the (phylogenetic investigation of communities by reconstruction of unobserved states) PICRUSt software package (v1.0.0) to predict gene contents and metagenomic functional information [49]. The statistical STAMP [50] was used to do the statistical analysis, and significant pathways (*p*-value 0.05, CI 99 percent) were exported and utilized to construct the results.

### 2.7. Statistical Analysis

The analysis by group of the continuous variables was performed using a Mann–Whitney–Wilcoxon test, while the association between two categorical variables was performed using a chi-square test (Table 1). A machine learning technique called L0L2-regularized regression [51], implemented in the R package L0Learn, was applied for the analysis of multivariate data. This method allows one to estimate the coefficients of regression and to select the best subset of variables in one single procedure. The unimportant variables are automatically estimated by zero. In addition, we used the R Package randomForest [52] to run a random forest regression to facilitate meaningful comparisons of predictor variables.

## 3. Results

A total of 112 subjects between 4 and 14 years old attending the Sidra Pediatric clinics and Emergency department was assessed for eligibility and consented to participate in the study. Subjects were excluded if they had chronic diseases, took antibiotics in the last month or were on vitamin D supplementation for the last six months before enrolment, failed to provide blood samples, and/or if their serum vitamin D status was unavailable. A total of 88 subjects met the above criteria (Table 1), but only 64 provided a stool sample and were included for gut microbiota profiling.

Based on the above information about vitamin D classification in the methods section and taking serum 25(OH)D of 50 nmol/L (20 ng/mL) as a point of reference, our data suggest 69% of the subjects fell into the vitamin D deficient category (Table 1), with a mean 25(OH)D of 36.13 nmol/L. There was a fair representation of male and female subjects in both the deficient and non-deficient cohorts. Most of the subjects belonging to both the deficient (75%) and non-deficient (59%) groups were of Arab origin (Appendix A). Average daily exposure to sun was above half hour in 82% of the deficient and 96% of the non-deficient subjects. After carrying out the group analysis, sun exposure was determined as a significant predictor variable.

### 3.1. Gut Microbial Composition and Diversity Are Altered in Vitamin D Deficient Children

The analysis of 16S rRNA gene sequencing data of stool samples (*n* = 63, one sample did not yield any PCR product) showed a significant decrease in bacterial diversity at the genus level in deficient (D) subjects compared to the non-deficient (ND) subjects as measured by the alpha diversity indices, observed species (*p*_Observed_ = 0.039), and Chao1 (*p*_Chao1_ = 0.047) (Figure 2a). Shannon and Simpson metrics for alpha diversity were borderline and failed to reach the significance level (*p*_Shannon_ = 0.052, *p*_Simpson_ = 0.057). The CCA profile based on the Bray–Curtis distance (Figure 2b) indicated that the two groups had significantly different microbiota community structures (*p* = 9 × 10^−3^). Interestingly, lower α-diversity and richness together with significantly different β-diversity as observed in the vitamin D deficient subjects as opposed to the non-deficient ones in our cohort were also seen in children with Crohn’s disease [53], inflammatory bowel disease [54], infections [55], diabetes [56], obesity [57], and others [58] when compared to healthy controls.

A total of 98–99% of the 16S rRNA gene sequences in both D and ND groups belonged to four major phyla: *Firmicutes, Bacteroidetes, Proteobacteria, and Actinobacteria* (Figure 2). In general, a healthy pediatric gut microbiota is dominated by *Firmicutes*, followed by *Bacteroidetes*, with these two major phyla representing more than 90% of the total gut bacteria [59,60]. Our results showed that the bacterial community composition was altered in the D group, with significantly greater abundance of *Bacteroidetes* (*p* = 0.019) and lower abundance of *Firmicutes* (Figure 3a). As such, the ratio of *Bacteroidetes* to *Firmicutes* (B/F *p* = 9.7× 10^−3^) was significantly higher in the D vs. ND group (Figure 3b). The B/F (or contrarily *Firmicutes/Bacteroidetes* (F/B)) ratio is known to play a role in maintaining intestinal homeostasis [61]. At the genus level, we observed that the D group had a significantly higher relative abundance of *Prevotella 9* (*p* = 6.6 × 10^−12^) and a significantly lower relative abundance of *Bacteroides* (*p* = 9.9 × 10^−6^) and *Alistipes* (*p* = 0.0069) compared with the ND group (Figure 3c,d and Appendix A). *Prevotella 9* and *Bacteroides* were the top two most abundant genera in our cohort (both belonging to phylum Bacteroidetes) and are also suggested to be two of the main enterotypes used to classify gut microbiota samples [62], often inferred by the *Bacteroides-to-Prevotella* Ratio (B/P) [63]. We found that the D group had a significantly lower B/P ratio compared to the ND group (Figure 3e). Considering the major taxonomic alteration at the phylum and genus levels, we then looked at the community shift at the species level; however, no significant shift was observed (data not shown). The abundance of several species of the genera *Bacteroides* and *Alistipes*, such as *Alistipes finegoldii*, *Alistipes* sp. *AL1*, *Alistipes* sp. *N15 MGS-157*, *Bacteroides caceae, Bacteroides eggerthii,* and *Bacteroides plebeius*, were higher in the ND subjects, whereas species such as *Prevotella dislens, Prevotella bivia, Bacteroides massillensis, Alistipes marseille,* and *Alistipes indistinctus* were higher in deficient subjects (Appendix A).

Therefore, our data indicate that the vitamin D deficient subjects had imbalanced gut microbial communities with a high B/F ratio, and a tendency towards a *Prevotella*-dominated enterotype.

### 3.2. Differential Functional Gut Microbiome Pathways in Children with Vitamin D Deficiency

To investigate the effect of vitamin D status on gut microbial function, phylogenetic investigation of communities by reconstruction of unobserved states (PICRUSt) was performed to predict functional abundances based on the 16S marker gene sequences. This meta-analysis identified 81 pathways that were significantly different between the two groups (Appendix A). In deficient subjects, 57 differential pathways were more abundant than that in non-deficient controls. Notably, the gut microbiome of deficient subjects was enriched in the orthologs related to lipopolysaccharide (LPS) biosynthesis and LPS biosynthesis proteins along with pathways related to primary immunodeficiencies, type II diabetes mellitus, when compared with non-deficient subjects. On other hand pathways related to fatty acid metabolism, glycerolipid metabolism, valine, leucine, and isoleucine metabolism were enriched in the ND group (Figure 4). Among other notable differences, reduction in xylene/dioxin degradation and increased metabolism of drugs and xenobiotics was also observed in the deficient subjects.

### 3.3. SNP Selection and Genotyping

The allele frequencies of the SNPs genotyped in this study were comparable to the frequencies reported for the HapMap project [64]. The details of the selected genes and SNPs are provided in (Appendix A). Three SNPs (rs10877012 in the CYP27B1 gene, rs7041 in GC gene, and rs2882679 in GC) were excluded based on the HWE test (*p* < 0.05) (Appendix A). Upon the removal of SNPs that failed the quality control methods, 34 SNPs were used for further analysis. A chi-square test was used to test the association of SNP genotype frequencies with D/ND classification, and an ANOVA test was performed to test the association between genotypes and vitamin D level (Appendix A). The CT genotype (SNP rs12512631 in the GC gene) was found to be associated with low levels of serum 25(OH)D and was highly abundant in the deficient compared to non-deficient subjects (Figure 5a,b). The GC gene is 42.5 kb long with 13 exons and is found on chromosome 4q11–q13 [65]. Vitamin D actions are considerably facilitated by the GC gene, for the vitamin D binding protein (VDBP), which transports vitamin D metabolites to multiple sites of action [65]. Polymorphism in VDBP can result in varying affinity for the active vitamin D (1,25(OH)2 D) metabolite [65,66]. The SNP rs12512631 is in the downstream 3′ region of the GC gene and has been linked to circulating 25(OH)D levels [67]. Upon correlating the two groups of data, we found that the overall microbial abundance was significantly different between the three genotypes of several SNPs included in the study (Appendix A) along with the SNP of interest rs12512631 (Figure 5c); however, these differences were not observed in relation with other notable taxonomic alteration such as the B/F ratio, abundance of *Prevotella* or *Bacteroides*, or the diversity measures (Appendix A). We also performed regression analysis to investigate the effect of various variables on the vitamin D levels. Several variables in our study, notably age, ethnicity, B/F ratio, BMI z scores, sun exposure, dairy consumption, and rs12512631 genotypes, were found to be highly predictive of vitamin D levels. The random forest test was conducted to determine variables of importance, and BMI z scores, B/F ratio, and age were among the most predictive variables of vitamin D levels (Figure 6). Similarly, in the multivariable regression (Appendix A), age, B/F ratio, exposure to sunlight, and BMI z scores were found to be strongest predictors of circulating 25(OH)D levels among other variables.

## 4. Discussion

Despite ample sunshine, approximately 80% of the population throughout the GCC region, including Qatar, suffers from vitamin D deficiency [9]. However, the information about the vitamin D status in the pediatric populations from Qatar remains scarce. High levels of deficiency were observed in a study conducted among the young adults in Qatar, mostly in the age group of 11–16 years, and among these, vitamin D deficient children had significantly higher rates of rickets, fractures, gastroenteritis, and delayed milestones [19]. Our study included children aged 4 to 14 years old, and we observed a high prevalence of vitamin D insufficiency/deficiency, with nearly 97 percent of the study participants demonstrating serum 25(OH)D levels below the sufficient level (75 nmol/L) [32].

Studies have shown associations between low vitamin D serum levels and conditions that may predispose patients to chronic diseases, such as obesity [68], metabolic syndrome [69], and cardiometabolic diseases [70]. Inadequate vitamin D levels during childhood and adolescence has also been linked to an increased prevalence of respiratory tract infections [71,72,73] and pediatric autoimmune-related disease [74]. This shows that vitamin D inadequacy in childhood may have a role in the progression of a variety of disorders and could help explain why hypertension, metabolic syndrome, hyperglycemia, obesity, diabetes, and cardiovascular disease are all on the rise in Qatar.

The 25(OH)D concentration has high heritability (28–80%) [14]. GWAS studies have identified several common genetic variants in genes related to the vitamin D pathway, the metabolism that influences 25(OH)D concentrations, and the risk of insufficiency [37]. Vitamin D-binding protein (VDBP) is the primary carrier of vitamin D metabolites in the blood, including 25(OH)D and 1,25(OH)2D, and is encoded by a group-specific component (GC) located on chromosome 4 (4 q11–13) [75]. VDBP affects the bioavailability of vitamin D metabolites by transporting them to the target tissues and thus may also influence disease risk associated with these metabolites [76]. GC variants have been linked to susceptibility to various diseases in the past [77], and recent GWAS studies have shown that allelic variation in GC contributes to changes in affinity of VDBP for the vitamin D metabolites as well as the serum concentrations of 25(OH)D [37,76,78,79]. Our data suggest that the CT genotype of rs12512631 (GC gene) was significantly associated with low levels of vitamin D in children and was highly prevalent in the vitamin D deficient subjects. Previous studies have shown that maternal rs12512631 (GC gene) genotypes had a significant impact on the association between 25(OH)D and the infant’s birth weight [80]. Additionally, rs12512631 was found to be linked to 25(OH)D levels in both children and adults [14], with a stronger impact in younger than older adults [81]. The reason behind the age-dependent association needs to be elucidated; lifetime environmental and/or body composition changes might also contribute to explain such a phenomenon. Studies have also highlighted the physiological importance of rs12512631 by demonstrating its associations with melanoma and prostate and colorectal cancers [65,82,83]. The rs12512631 SNP is found in the 3′ untranslated region of GC [65]. While the exact consequences of rs12512631 on the function of VDBP are unknown, it could alter the gene expression via disruption of miRNA binding, post-transcriptional processing, and 3′-cleavage/polyadenylation; however, large-scale genomic studies with a bigger cohort size are required to examine the full functional impact of this SNP in pediatric subjects.

As for the other SNPs, it should be highlighted that there might be other loci, different from the ones tested in our study, that might have a role in the association with vitamin D in the pediatric population of Qatar. Additionally, it is possible that the sample size of our cohort did not carry enough power to detect association.

The gut microbiota is a diverse community of microorganisms that exist in symbiotic relationship with their host and perform essential functions such as digestion, intestinal homeostasis, and the maturation and education of the immune system [83]. Like any ecosystem, a healthy, immune resilient, and stable gut relies on high microbiota richness and biodiversity [84]. Low diversity on other hand is commonly found is several disease states such as obesity [85,86], type 2 diabetes [87], and IBD [88]. The gut microbiota diversity of healthy pre-adolescent children aged 6–12 years has been shown to be substantially higher than that of healthy adults living in the same environment [89]. Our vitamin D deficient children’s gut microbiome showed a considerably low alpha diversity (observed species and Chao 1). The same low-diversity dysbiotic states were observed in our recent interventional study with vitamin D deficient adult subjects, where supplementation with vitamin D resulted in an increase in both the richness and diversity of gut microbiota [21]. The presence of VDRs in the gastrointestinal tract could be one of the methods by which vitamin D influences microbial composition and diversity, particularly through immune response control, modulation, and preservation of gut epithelial integrity [90]. Antimicrobial peptides such as cathelicidin and defensin are also produced by VDRs and (1,25(OH)2D), the active bioavailable form of vitamin D [91,92]. These antimicrobial peptides are important for maintaining microbial equilibrium.

Vitamin D is a fat-soluble molecule, and in order for it to be absorbed, it must be made water-soluble in the intestines [93]. This is accomplished by emulsification of vitamin D in the intestinal lumen, through the action of bile acids, forming small droplets which are incorporated into micelles—complex aggregates facilitating its absorption by the intestinal cells [94]. Without proper functioning of this mechanism, the relative bioavailability of vitamin D will be low. A favorable gut microbial environment produces beneficial metabolites such as SCFAs, bile acids driving increased micelle formation/assembly and regulation of intestinal barrier integrity, which allows the proper absorption of vitamin D. On the other hand, alteration in human bile acid pool composition induced by changes in the gut microbiota may influence the absorption and bioavailability of vitamin D. This suggests a central role for the gut microbiome and its metabolites in the mechanism of vitamin D absorption. However, there is a paucity of human studies in this field and there is a great need for research to elucidate the exact mechanisms to support this hypothesis.

About 90% of the intestinal bacteria can be assigned towards two major bacterial phyla, the “Firmicutes” and “Bacteroidetes”. Composition-wise, Firmicutes (~60%) generally overtakes Bacteroidetes (~20%) in the healthy adult human gut [95]. In comparison to adults, the average healthy child’s gut community has significantly more Firmicutes and less Bacteroidetes [89]. In our study, vitamin D deficient children exhibited an overabundance of Bacteroidetes, resulting in a significantly higher Bacteroidetes/Firmicutes (B/F) ratio compared to non-deficient children. The B/F ratio is commonly considered to have a key role in maintaining proper intestinal homeostasis, and disbalance in the ratio is often regarded as a general determinant for gut dysbiosis [61]. Microbial dysbiosis associated with inflammatory gastrointestinal diseases such as IBD is characterized by an increase in the abundance of the phylum Bacteroidetes and a decrease in Firmicutes, resulting in a higher B/F ratio [61,96,97]. Interestingly vitamin D deficiency is highly prevalent in IBD patients (as high as 90% in some cases) and was found to be significantly associated with disease activity [98]. In addition, many studies have confirmed the close relation between vitamin D deficiency and the development of insulin dysregulation/T2DM, increased length of respiratory infections, and mortality in patients with common immunodeficiencies [99,100,101]. Dysbiosis of the gut microbiota caused by vitamin D deficiency could be a possible reason for increased vulnerability to inflammatory and immune-mediated illnesses. Bacteroidetes are gram-negative bacteria that can induce activation of macrophages via the LPS present on their surface, triggering proinflammatory cascade and systemic inflammation that can cause infection or diseases under certain conditions [102]. In support of this observation, our PICRUSt analysis revealed that the gut microbiome involved biosynthesis pathways related to LPS biosynthesis that were elevated in vitamin D deficient children.

Wu *et al.* classified fecal communities into two enterotypes based on the abundance of *Bacteroides* and *Prevotella* and discovered that vitamin D intake was strongly positively associated with the *Bacteroides* enterotype and negatively to the abundance of the *Prevotella* enterotype [29]. Subsequently, studies have shown that vitamin D supplementation resulted in an increased abundance of *Bacteroides* and a decrease in *Prevotella* in both human and animals [21,103]. In line with the above data, we found overabundance of *Prevotella* enterotypes in the gut of vitamin D deficient children, resulting in a significantly higher *Prevotella* to *Bacteroides* ratio. Individual genes, the external environment, and eating habits all influence the prevalence of one enterotype over another, resulting in uniquely distinct host microbiomes [104]. These finding could indeed be very promising in the field of personalized medicine, as individual stratification based on these two microbial enterotypes (i.e., the dominance of *Prevotella* or *Bacteroides*) could help predict responses to dietary supplements or medications [105,106,107].

A high prevalence of vitamin D deficiency/insufficiency in the subjects enrolled in the study suggests the need for urgent preventative actions. Several predictors of vitamin D levels have been reported in the past (e.g., age [108], sex, BMI, skin color and protection, vitamin D supplementation, season, latitude) [4,109,110,111,112] that allow for the detection of populations at risk, which may benefit from attentive prevention. The strongest predictors of vitamin D levels in our study were BMI z scores, age, and the B/F ratio. Evidence suggests that obese children are more prone to vitamin D deficiency than non-obese children [113], and the degree of deficiency is directly related to adiposity [113]. Our univariate analysis showed that vitamin D deficient children showed higher BMI compared to non-deficient (Table 1, Appendix A), suggesting that overweight children are at significantly greater risk of vitamin D inadequacy. Additionally, the multivariate regression analysis demonstrated that BMI z scores were inversely related to vitamin D levels; these data were consistent with numerous studies demonstrating that BMI z scores are strong predictors of vitamin D levels, especially in children [114,115,116,117,118,119]. This reduction in the levels of serum 25(OH)D coinciding with an increase in BMI z scores may be due to volumetric dilution of vitamin D in the large adipose stores or excess sequestration of vitamin D in fat, leading to decreased bioavailability [115]. Earlier research showed that subcutaneous synthesis of vitamin D declines with age [120]; our study shows a negative association of age with vitamin D levels, suggesting that older children are more susceptible to vitamin D deficiency.

Milk and milk products contribute nearly half of the dietary vitamin D intake in many countries [121,122]. Our data indicate that dairy consumption is a significant predictor of vitamin D status, and people who consumed dairy are less likely to suffer from low vitamin D levels. The duration of exposure to sunlight played an important role, with children who were exposed to sunlight for more than 1 h daily showing a positive association with higher vitamin D levels, whereas children who received less than 30 min of sunlight showed a negative association. In addition, children with Arab ethnicities and those with family history of vitamin D deficiency were more likely to suffer from low vitamin D levels.

Interestingly, we also found significant predictive potential of B/F ratio and rs12512631 genotypes on the vitamin D status; these are novel findings and considering the importance of these variables, one may consider genotyping for the risk allele and to predict the population prone to vitamin D deficiency, gut microbiota targeted therapeutic intervention such as the use of probiotics/prebiotics in combination with higher consumption of dairy products can be recommended for the risk population. However, replication studies are warranted to confirm our findings in cohorts of bigger sizes.

Altogether, our results confirm that the gut microbial diversity, B/F ratio, and *Prevotella* driven gut enterotypes appear to be general features distinguishing deficient and non-deficient gut microbiota in pediatric subjects. A myriad of factors may impact microbial communities, including host genetics. Thus, based on the promising results, we expanded our efforts to examine the interactions between genetic variation and the gut microbiota. We were able to identify significant associations between specific SNPs and the overall microbiome composition. All the three genotypes of the SNP of interest, rs12512631, were identified to be significantly associated with vitamin D levels and showed a significant difference in the overall microbial abundance; however, no relation was observed with other biomarkers such as B/F ratio, diversity indices, or P/B ratio. We believe the reason for this could be the smaller cohort size, and studies with larger cohorts are needed to delineate the association.

Our findings underscore the need to consider host genetics and baseline gut microbiota makeup in interpreting vitamin D status and designing better personalized strategies for therapeutic interventions.

## Figures and Tables

**Figure 1 biomedicines-10-00278-f001:**
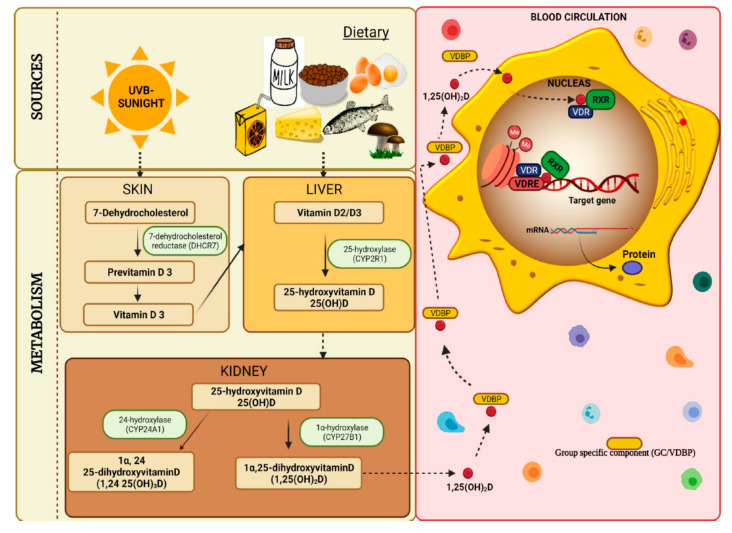
The chosen candidate genes’ involvement in the vitamin D cascade are displayed. Vitamin D is mostly obtained through sunlight or food sources in humans. UVB light from the sun penetrates the skin and converts 7-dehydrocholesterol (7DHC) to pre-vitamin D, which is rapidly transformed to vitamin D. *DHCR7/NADSYN1* removes 7DHC from the vitamin D pathway. D is hydroxylated in the liver to 25(OH)D3, primarily by *CYP2R1*. Following that, 25(OH)D is delivered to the kidney by vitamin D binding protein (VDBP), which is encoded by *GC*, where it is converted by *CYP27B1* to its active form, 1,25(OH)2D. Finally, *CYP24A1* catabolizes both 25(OH)D and 1,25(OH)2D to calcitroic acid, which is physiologically inactive and water soluble. The active form of vitamin D bound to VDBP is delivered in blood to its target cells, where it acts as a ligand for the vitamin D receptor (*VDR*), a nuclear transcription factor that regulates transcription and translation of messenger RNA, leading to the synthesis of vitamin D-dependent proteins.

**Figure 2 biomedicines-10-00278-f002:**
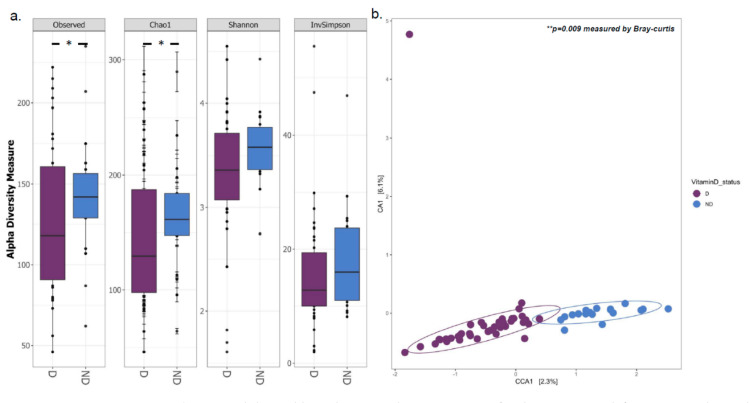
Alpha and beta diversity plot comparisons for the vitamin D deficient (D) and non-deficient (ND) groups. (**a**) Observed species, Chao1, Shannon, and Inverse Simpson boxplots representing the alpha-diversity indices. The horizontal line inside the box indicates the median, while the boxes reflect the interquartile range (IQR) between the first and third quartiles (25th and 75th percentiles, respectively). Whiskers reflect the lowest and greatest values from the first and third quartiles that are within 1.5 times the IQR, respectively. The Wilcoxon test with FDR–Bonferroni corrected *p* values was used to determine statistical significance. * *p* < 0.05; (**b**) CCA plot showing the beta diversity measure ** *p* < 0.01 deep purple: deficient samples, blue: non-deficient. Each dot represents an individual sample. The figure was generated using RStudio v 1.4 with R v 4.1.

**Figure 3 biomedicines-10-00278-f003:**
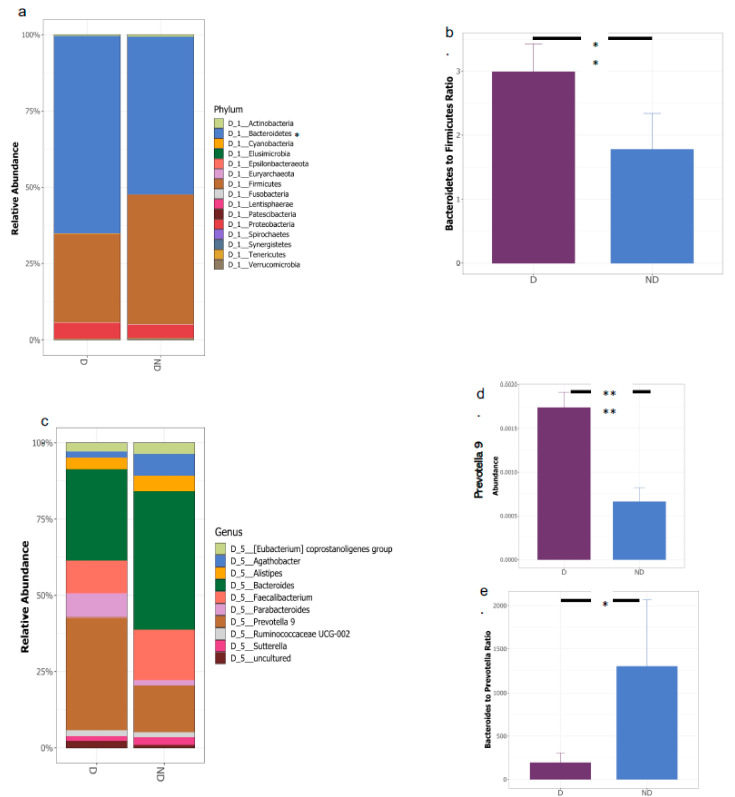
Comparison of the gut microbiota composition in vitamin D deficient (D) and non-deficient (ND) groups at the phylum level. (**a**) Relative abundance of different bacterial phyla in the deficient and non-deficient groups. The relative abundance of the major bacterial phyla, Firmicutes and Bacteroidetes, showed variation between the two groups. The abundance of Phylum Bacteroidetes was significantly elevated in the D group as compared to ND (Wilcoxon test with false discovery rate (FDR)–Bonferroni corrected, * *p* < 0.05). The figure was generated using RStudio v 1.4 with R v 4.1. (**b**) Comparison of the ratio of Bacteroidetes to Firmicutes in vitamin D deficient (D) and non-deficient (ND) groups (Wilcoxon test with false discovery rate (FDR)–Bonferroni corrected, * *p* < 0.05). The figure was generated using RStudio v 1.4 with R v 4.1. Comparison of the gut microbiota composition in vitamin D deficient (D) and non-deficient (ND) groups at the genus level. (**c**) Relative abundance of different bacterial genus in the deficient and non-deficient groups. The relative abundance of the major bacterial genera, *Prevotella* and *Bacteroides*, showed variation between the two groups. (**d**) Abundance of genus *Prevotella 9* was significantly elevated in the D group as compared to ND. (**e**) Comparison of the ratio of *Bacteroides* to *Prevotella* in vitamin D deficient (D) and non-deficient groups (ND) groups (Wilcoxon test with false discovery rate (FDR)–Bonferroni corrected, * *p* < 0.05 and ** *p* < 0.01; **** *p* < 0.0001. The figure was generated using RStudio v 1.4 with R v 4.1.

**Figure 4 biomedicines-10-00278-f004:**
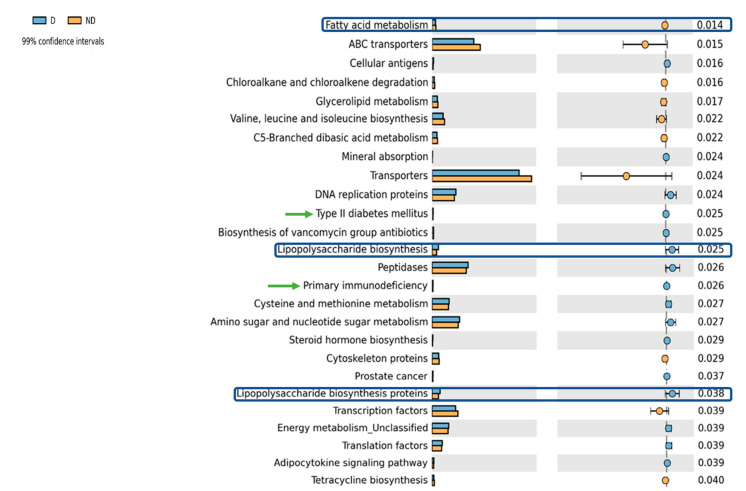
PICRUSt was used to infer gut microbiota functions using the 16S rRNA gene sequence data from vitamin D deficient (D) and non-deficient (ND) groups. The differences in projected functions of genes involved in fatty acid metabolism, as well as lipopolysaccharide biosynthesis and lipopolysaccharide biosynthesis proteins, are highlighted.

**Figure 5 biomedicines-10-00278-f005:**
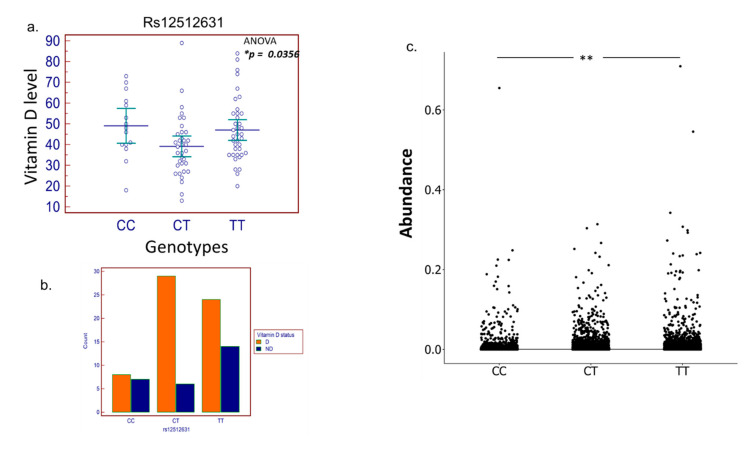
(**a**) The CT genotype (SNP rs12512631 in the GC gene) was associated with low levels of serum vitamin D and (**b**) was highly abundant in the deficient compared to non-deficient subjects (ANOVA * *p* < 0.05). (**c**) Overall microbial abundance was significantly different between the three genotypes of SNP rs12512631 (Kruskal–Wallis followed by post hoc Dunn’s test with false discovery rate (FDR)–Bonferroni corrected, ** *p* < 0.01).

**Figure 6 biomedicines-10-00278-f006:**
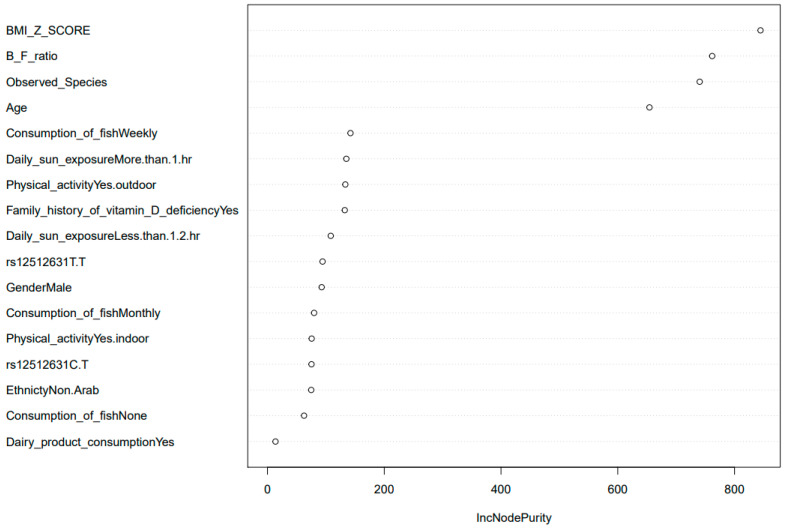
Random forest analysis used to determine the variables that had the largest contribution in determining the vitamin D levels. Inc Node Purity was used as a measure of “variable importance”.

**Table 1 biomedicines-10-00278-t001:** Baseline characteristic of the study participants.

	Vitamin D Status		
	Deficient (below 20 ng/mL or 50 nmol/L)	Non-Deficient (above 20 ng/mL or 50 nmol/L)	*p*-Values
Number	61	27	n/a
Age (years)	9.02 ± 3.23	8 ± 2.69	0.14
BMI, mean ± SD	19.4 ± 5.65	15.8 ± 4.08	0.049
BMI z-score, mean ± SD	0.357 ± 1.65	0.054 ± 1.73	0.54
25(OH)D levels	36.1 ± 8.52	62.6 ± 11.0	9.3 × 10^−14^
**Gender**, ***n*** (**%**)			
Male	31 (50.82%)	14 (51.85%)	1.00
Female	30 (49.18%)	13 (48.15%)	
**Ethnicity**, ***n*** (**%**)			
Arab	46 (75%)	16 (59%)	0.2
non-Arab	15(25%)	11 (40%)	
**Average Daily Exposure to Sun**			
Less than 1/2 h, *n* (%)	10 (16%)	1 (0.03%)	0.781
1/2 h to 1 h, *n* (%)	28 (45%)	22 (81.4%)	
More than 1 h, *n* (%)	23 (37%)	4 (14.8%)	
**Consumption of Fish**			
Daily, *n* (%)	2 (0.03%)	1 (0.03%)	0.7104
Weekly, *n* (%)	26 (42.6%)	15 (55%)	
Monthly, *n* (%)	22 (36%)	7 (25%)	
None, *n* (%)	11 (18%)	4 (14.8%)	
**Consumption of Dairy Products**, ***n*** (**%**)			
YES	58 (95%)	25 (92.5%)	1.00
NO	3 (4.9%)	2 (7.4%)	
**History of Vitamin D Deficiency**, ***n*** (**%**)			
YES	14 (22.95%)	8 (29.63%)	0.5619
NO	47 (77.05%)	19 (70.37)	

Abbreviation: BMI, body mass index. Chi-square test was used for categorical variables, and Mann–Whitney test for comparing continuous variables.

## Data Availability

The data presented in this study are available upon request from the corresponding author.

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
