# Peer review of "Tipping the Balance: Vitamin D Inadequacy in Children Impacts the Major Gut Bacterial Phyla"

_biomedicines, 2022, doi:10.3390/biomedicines10020278_

Round 1

Reviewer 1 Report

Singh and colleagues have performed a timely study in a pediatric/adolescent population in Qatar to investigate the relationship between Vitamin D status, genetic polymorphisms and the fecal microbiome. The strengths of this study are the targeted approaches and rationale, thoroughly described methods, and excellent presentation of results. The findings of the robust differences in microbiota between deficient and sufficient groups and the relationship observed between deficiency and the VDBP gene are very promising, and more insightful conclusions can and should be drawn from this data after further analysis.

Major comments

The authors conclude that host genetics and fecal microbiome should be used to interpret Vitamin D status in order to design better strategies for intervention, however the data collected should be further analyzed to generate a hypothesis to be tested in a larger cohort. Once between group comparisons in table 1 have been conducted, several measures will have predictive power of Vitamin D status. These variables (BMI z score, age, ethnicity, sun exposure, dairy consumption) should then be included with B/F ratio, observed species, GC genotype, and possibly other variables in a regression analysis with Vitamin D as the dependent variable. This is essential to understand the emphasis of therapeutic interventions. Hypothetically, you may conclude that CG genotype and BMI z score were the strongest predictors of Vitamin D status; thus, you may propose that CG genotyping be done and recommend more sun (physical activity outdoors) or dietary Vitamin D exposure for those with the risk allele. You may also use B/F ratio as the dependent variable and find that the CG risk allele predicts the discrepancy; thus loss of unbound Vitamin D may alter Bile Acid synthesis and microbiome structure.

Table 1 should provide between group analyses and percentages should be included for all variables. BMI should be reported as BMI z score given the range of ages of participants. Additionally, puberty status, especially in females, is a major factor that should be reported. Because non-Arab is a substantial percentage of the sample, the actual ethnicities of non-Arabs should be included in the text or potentially in the table. Was dairy consumption included in the dietary questionnaire? If so, this should be included because Vitamin D fortified dairy is a primary dietary source of Vitamin D.

Methods

When were data collected? Though Qatar is relatively close to the equator, would seasonality confound Vitamin D levels? No references were included for the questionnaire. Please include the questionnaire for reviewers so that reliability of the instrument(s) can be assessed. Genotyping is discussed and Vitamin D levels are reported but there is no documentation of phlebotomy (fasted?) or discussion of Vitamin D assessment methods.

Selection of Vitamin D threshold lines 189-199 should be included in methods, not results.

Selection of genes of interest lines 294 – 306 should be included in methods, not results.

Figure 6 should appear in methods as presented. Otherwise, p values could be reported for each gene analyzed, highlighting that CG was the only predictor in your study. Otherwise, The first table in Supplementary tables could replace the figure.

If DADA2 was used, the data that was output should have been in ASVs rather than OTUs.

The alpha diversity metric reported is observed species, not observed OTUs.

Results

Figures 2,3, and 4 should be combined in to one figure, omitting the bar charts and making notations in the abundance figures of significant differences.

Figure 7a – which D genotypes are significantly different? Figure 7c – what is on the y axis?

Discussion

Regression analysis should substantially bolster the breadth and depth of the discussion.

Reviewer 2 Report

The authors investigated the association between the vitamin D level and intestinal microbiota as well as host genetic factor in children of the Middle East where there is a high prevalence of vitamin D deficiency. They showed that vitamin D deficient children had a low diversity of gut microbiota, which may be associated with a specific single nucleotide polymorphism (SNP) in the GC gene coding for the vitamin D binding protein. This study is interesting as it can expand our knowledge in this field. However, I have some comments and suggestions for further improvement.

  1. Could you please consider describing possible mechanisms on how vitamin D deficiency can lead to poor gut microbial diversity? Could a reduction in the gut microbial diversity result in low vitamin D absorption from the gut? If so, please consider discussing this point, too.
  2. Is an allelic frequency of the CT genotype in rs12512316 higher in the Middle East than those in other regions? This information might help us understand the reason for a high prevalence of vitamin D deficiency in the Middle East. Could you please consider discussing this point?
  3. Please define the abbreviation 25(OH)D for readers who may be novice in this field (line 22). Please use a uniform abbreviation for this metabolite of vitamin D throughout the manuscript (e.g., 25OHD in line 25 and 359, 25-OH-D in line 357).
  4. Please change “a mean vitamin D” in line 201 to “a mean 25(OH)D.”
  5. Some commas appear to be missing. For example, between “gastroenteritis” and “fractures” in line 57, between “obesity [67]” and “metabolic syndrome” in line 350.
  6. Please insert a space between “75” and “nM” in line 46 and between “tract” and “(GIT).”
  7. Please delete the unnecessary underlining in lines 133, 377, and 393.

Round 2

Reviewer 1 Report

I was happy to see that the authors worked to address all of my questions and concerns in their revision, however, it appears a biostatistician was not included in the additional analyses because they were not properly designed, adequately assessed, or interpreted. Table 1 was not analyzed with the correct tests, nor were the regression analyses properly conducted. The results of the regression actually point to a specific hypothesis that the authors should discuss as a means for preventing Vitamin D deficiency that should be further investigated. Please add a biostatistician to your team to improve the quality of analysis as you have a truly well-designed study and exciting findings to publish.

Author Response

I was happy to see that the authors worked to address all of my questions and concerns in their revision, however, it appears a biostatistician was not included in the additional analyses because they were not properly designed, adequately assessed, or interpreted. Table 1 was not analyzed with the correct tests, nor were the regression analyses properly conducted. The results of the regression actually point to a specific hypothesis that the authors should discuss as a means for preventing Vitamin D deficiency that should be further investigated. Please add a biostatistician to your team to improve the quality of analysis as you have a truly well-designed study and exciting findings to publish.

Response: We thank the Reviewer for highlighting the need for the above improvements. As suggested by the reviewer we have included biostatistician to our team, who helped us conduct additional analysis and interpretation of the results. Table 1 has been reanalyzed using Mann–Whitney–Wilcoxon test with group of the continuous variables, while the association between two categorical variables was performed using a Chi-square test. The result of univariate analysis of the impact of significant variable on vitamin D levels have also been included as suppl. figure S6.

We then performed regression analysis using machine learning technique called L0L2-regularized regression [1], for the analysis of multivariate data. This method allowed us to estimate the coefficients of regression and to select the best subset of variables in one single procedure. The results of this analysis have been included in suppl. table 7.  In addition, we used the R Package randomForest  to run a Random Forest regression to determine the importance of each variable in the prediction of Vitamin D level. Please refer to figure6.

Based on the results from the two analysis we found that several common variables have the predictive power of Vitamin D status, such as age, BMI z scores, B/F ratio, and well as the rs12512631 genotypes. We have discussed these results in detail in the discussion section. Please refer to lines 750-783.
